# A Novel Fluorescent Probe for Determination of pH and Viscosity Based on a Highly Water-Soluble 1,8-Naphthalimide Rotor

**DOI:** 10.3390/molecules27217556

**Published:** 2022-11-04

**Authors:** Ventsislav V. Bakov, Nikolai I. Georgiev, Vladimir B. Bojinov

**Affiliations:** 1Department of Organic Synthesis, University of Chemical Technology and Metallurgy, 8 Kliment Ohridsky Str., 1756 Sofia, Bulgaria; 2Bulgarian Academy of Sciences, 1040 Sofia, Bulgaria

**Keywords:** 1,8-naphthalimide, twisted intramolecular charge transfer (TICT), fluorescence, pH, viscosity

## Abstract

A novel highly water-soluble 1,8-naphthalimide with pH and viscosity-sensing fluorescence was synthesized and investigated. The synthesized compound was designed as a molecular device in which a molecular rotor and molecular “off-on” switcher were integrated. In order to obtain a TICT driven molecular motion at C-4 position of the 1,8-naphthalimide fluorophore, a 4-methylpiperazinyl fragment was introduced. The molecular motion was confirmed after photophysical investigation in solvents with different viscosity; furthermore, the fluorescence-sensing properties of the examined compound were investigated in 100% aqueous medium and it was found that it could be used as an efficient fluorescent probe for pH. Due to the non-emissive deexcitation nature of the TICT fluorophore, the novel system showed low yellow–green emission, which represented “power-on”/“rotor-on” state. The protonation of the methylpiperazine amine destabilized the TICT process, which was accompanied by fluorescence enhancement indicating a “power-on”/“rotor-off” state of the system. The results obtained clearly illustrated the great potential of the synthesized compound to serve as pH- and viscosity-sensing material in aqueous solution.

## 1. Introduction

Due to the rapid progress in the fields of chemosensing materials and nanotechnology, the design and synthesis of molecular switches have received considerable attention in recent years [1,2,3,4]. A wide range of different architectures applied as various molecular devices [5,6,7,8,9,10], object-coding [11], chemical-sensing [12,13], data-storage [14,15], drug-delivery [16,17], and drug-activation [18,19] systems have been obtained. Among them, major attention has been paid to fluorescence sensors and probes due to the several advantages, such as cheap equipment, high sensitivity, immediate response, and great spectral resolution. In addition, they have a small, safe, and indestructible signaling nature which allows their practical application even in real-time imagining of living organisms [20,21,22,23,24].

The lack of water-solubility is the main disadvantage of fluorescent probes, which seriously restricts their usage. Currently, chemical analysis in water solutions, in the absence of organic solvents, was preferable due to the use of environmentally friendly media with lower cytotoxicity and higher biocompatibility, which is, especially, suitable for bioimagining purposes; therefore, the design and synthesis of highly water-soluble fluorescent probes has increasingly attracted considerable interest [25,26,27,28,29]. This encouraged us to prepare and investigate the chemosensing properties of a novel fluorescent probe with high solubility in 100% water media. The compound presented in Figure 1 under study is a TICT (twisted intramolecular charge transfer) molecular rotor with fluorescence-sensing properties focusing on pH and viscosity. The rapid analysis of pH and viscosity plays a critical role in large areas of industrial production, food processing, and environmental monitoring [30,31,32,33,34,35,36,37]; also, the abnormal values of intracellular pH and viscosity could be associated with several diseases, such as atherosclerosis, Alzheimer’s disease, diabetes, and cancer [38,39,40,41,42]; hence, the synthesis of fluorescent probes for both pH and viscosity currently could be of significant importance. In order to simultaneously achieve pH- and viscosity-switchable fluorescence, the novel probe was based on a TICT molecular rotor platform. It is well known that deexcitation from the TICT state is non-radiative or shows batochromically shifted (lower energy) emission than normal fluorescence which was successfully utilized for the imaging of viscosity in biological objects [43,44]; furthermore, the TICT process is microenvironmentally dependable and can be easily switched at different pHs [45].

## 2. Results and Discussion

### 2.1. Design and Synthesis

The compound under study (**3**) was designed as a fluorescence-sensing TICT rotor based on a 4-piperazinyl-1,8-naphthalimide architecture. The 1,8-naphthalimide fluorophore was chosen as a fluorogenic unit due to its bright fluorescence, large stokes shift, and high photo and chemical stabilities [46,47,48]. The *N*-methylpiperazine fragment was bound to the C-4 position of the 1,8-naphthalimide unit in probe **3** to achieve TICT molecular motion. It is well known that the presence of dialkylamines such as *N*-methylbuthylamine, morpholine, piperidine, and piperazine at C-4 position resulted in a TICT process in the fluorophore-excited state and they could be used as fluorescent probes for viscosity [49,50,51,52]; additionally, the protonation of *N*-methylpiperazine amine destabilized the irradiative TICT process, thus preventing rotation and allowing significant emission increase which was widely utilized in the pH-probe design [9,53,54]. Furthermore, the presence of different amines at *N*-position of the 1,8-naphthaimides resulted in a significant increase in their water solubility [25,55,56]. This was our motive to introduce a primary amino group in position *N* of the novel probe.

The examined compound **3** was easily synthesized from available sources in two steps according Figure 2.

First, hydrazine monohydrate was condensed with 4-cloro-1,8-naphthalic anhydride using equimolar amounts in methanol solution under reflux; then, the chlorine in intermediate **2** was, subsequently, substituted with *N*-methylpiperazine in boiling DMF solution for 5 h to afford final probe **3** as yellow crystals.

### 2.2. Chemosensing Properties of Probe **3**

#### 2.2.1. pH-Sensing Properties

The synthesized compound **3** was designed as a fluorescent probe with high water solubility. This was the reason to study its photophysical properties in 100% water solution (in the absence of organic solvents). It was found that in acid and neutral media probe **3** showed an absorption band in a range of 300–500 nm with maximum at 394 nm that was slightly red, shifting under alkaline conditions to 406 nm (Figure 1). Such behavior is common for 4-amino-1,8-naphthalimides containing aminoalkylamines at position C-4 and easy could be rationalized according to the internal charge transfer (ICT) occurring in the excited state of these chromophoric systems. The light absorption of 1,8-naphthaimides results in a charge transfer from the C-4 electron-donating group to the electron-accepting carbonyls, which efficiency determines the basic photophysical properties of the molecule [57,58,59]. For a difference, in neutral and acid media the methylpiperazine nitrogen is in its protonated form and exerts a weak charge repulsion on the 4-amino moiety directly attached to the ICT cromophoric system in probe **3**. This decreased the ICT efficiency and slightly shifted the observed absorption band towards higher energy wavelengths.

In alkaline media probe **3** showed a very low fluorescence emission in the range of 450–650 nm, with maximal value at 550 nm (Figure 2A); however, the protonation of the methylpiperazine amine after addition of hydrochloric acid gradually increased the fluorescence output of **3** and blue shifted its maximum to 530 nm. The calculated quantum yield of fluorescence was *Φ*_F_ = 0.001 at pH 12 and *Φ*_F_ = 0.14 at pH 4.

The observed fluorescent enhancement in acid media was expectable and, in recent works, it was attributed to the fact that the ICT in 4-methylpiperzinyl-1,8-naphthalimides undergoes a twisting process (TICT) with nonradiative deexcitation nature [51]. The protonated methylpiperazine nitrogen generates a positively charged cation that leads to electrostatic destabilization of the TICT state, thus reducing its deexcitation channel. As a result, a bright fluorescence was observed. From the fluorescent changes at 530 nm as a function of pH, a well pronounced S-shaped (sigmoidal Boltzmann fit, R^2^ = 0.9977) titration plot was observed, suggesting a simple thermodynamic equilibrium (Figure 2B).

It was found that the reversible pH-switching process appeared in pH window 6–9, giving a p*K*_a_ value of 7.69 ± 0.05 according to the Hendersen-Hasselbalch Equation (1), which matches the previous reported p*K*_a_ value of 4-methylpiperzinyl-1,8-naphthalimide derivatives [51].
(1)$$\mathrm{pH}=pK_{a}+\log\frac{\left(I_{\max}-I\right)}{\left(I-I_{\min}\right)}$$
where *I*_max_ and *I*_min_ are the maximum and minimum fluorescence intensity, respectively, and *I* is the fluorescence intensity at the corresponding pH value.

Furthermore, the difference in the TICT and normal states showed that the deexcitation of **3** could be used as a molecular rotor with power indicator at molecular level giving us information about the presence or lack of motion in the TICT system. The non-emissive TICT deexcitation nature of **3** shows very low yellow-green emission, which represents the “power-on”/“rotor-on” state of the system. The protonation of the methylpiperazine amine destabilizes the TICT process, thus enhancing the fluorescence intensity of probe **3,** indicating a “power-on”/“rotor-off” state of the system (Figure 3).

The effects of the most common cations and anions (Co^2+^, Cu^2+^, Fe^3+^, Ni^2+^, Pb^2+^, Cd^2+^, Zn^2+^, Hg^2+^, Cl^−^, NO_3_^−^, SO_4_^2−^, HSO_4_^−^, CO_3_^2−^, CH_3_COO^−^, Br^−^, NO_2_^−^, SO_3_^2−^, PO_4_^3−^, and F^−^) on the fluorescent emission of **3** were tested as potential analytes or interferents. The study was performed in aqueous media at pH 7.2 (10 µM HEPES) and pH 8 (10 µM Tris-HCl). In both cases, the tested ions (10^−5^ mol/L and 10^−4^ mol/L) caused only a minor quenching (below 10%) of the probes’ (10^−5^ mol/L) fluorescence intensity; also, it was found that the studied probe **3** could be transferred between “off” and “on” states reversibly at least nine times without changes in the fluorescence intensity in both acid and alkaline media (Figure 3). These results clearly showed that compound **3** is stable in a wide pH range and could be used as a selective and efficient platform for rapid determination of pH values in aqueous solutions.

#### 2.2.2. Viscosity-Sensing Properties

The fluorescence properties of **3** were investigated in solvents with different viscosities in order to confirm the existence of TICT molecular motion in the examined compound. The higher viscosity seriously restricts the intramolecular bond rotation and prevents the TICT process. The influence of viscosity on the fluorescence spectra of **3** was defined after measurements in glycol, glycerol, and glycol/glycerol mixtures (Figure 4). As can be seen from Figure 3, the probe **3** showed viscosity-sensing fluorescence intensity increasing in a high viscous solution. A double logarithmic scale of the fluorescent intensity at 530 nm of compound **3** and solvent viscosity (Figure 4, Inset) showed a relationship with good linearity (R^2^ = 0.9843) which is typical for TICT rotors [36,37].

Glycerol has lower p*K*_a_ value than glycol and the acidity of the glycerol/glycol mixture could be a major reason for the observed fluorescence changes. In order to reject this theory, the fluorescent spectra of **3** were measured in glycol, buffered glycol containing 10 µM Tris-HCl (pH 8), glycerol, and buffered glycerol containing Tris-HCl (pH 8). The results presented in Figure 4B show that the presence of buffer solution did not lead to a significant result which clearly illustrated that the changes in fluorescence intensity of **3** mainly were induced due to the different viscosity. The results suggested the existence of fluorescence quenching in **3** with the nature of the TICT deexcitation path making it a promising fluorescent probe for viscosity.

#### 2.2.3. Molecular Logic

As a whole, the above fluorescence-sensing properties of probe **3** logically could be summarized in an OR molecular logic gate (Figure 5). In low-viscosity solution (glycol, coded in binary as 0) and in the absence of protons (HCl, coded in binary as 0), **3** showed a very low fluorescence output which was coded in binary as 0; however, in solution with high viscosity (glycerol, coded in binary as 1) and in the absence of protons (hydrochloric acid, coded in binary as 0), probe **3** fluorescence was higher (coded in binary as 1) due to the hindered TICT quenching effect. In the presence of protons (HCl 10^−6^ mol/L, coded in binary as 1), due to the destabilization of the TICT excited state, probe **3** showed viscosity independent of high fluorescence output (coded in binary as 1). In other words, the novel probe shows high fluorescence at high viscosity or high acidity, or both high viscosity and high acidity. This behavior correlated very well with an OR logic gate. Obviously, the novel probe **3** could be used as OR molecular logic gate, using viscosity and protons as inputs and fluorescence emission as output.

In addition, it should be pointed out that in the presence of protons in glycerol the observed emission was higher than in glycerol without acid. This could have been the result of a possible photoinduced electron transfer (PET) from tertiary piperazine amine to the excited fluorophore, which quenched the fluorescence in glycerol. The protonation of the tertiary amine disallowed the PET quenching, thus, the addition of acid in glycerol solution resulted in a brighter fluorescence [51].

## 3. Materials and Methods

### 3.1. Materials

The starting reagents 4-chloro-1,8-naphthalic anhydride, hydrazine monohydrate, and *N*-methylpiperazine were used as commercial products (Sigma-Aldrich Co., St. Louis, MO, USA and Fisher Scientific, Waltham, MA, USA) without purification. All solvents used in the synthetic procedures and in the photophysical investigation (Sigma-Aldrich Co., St. Louis, MO, USA and Fisher Scientific, Waltham, MA, USA) were pure or of spectroscopy grade. As sources of metal cations, Zn(NO_3_)_2_, Cu(NO_3_)_2_, Ni(NO_3_)_2_, Co(NO_3_)_2_, Pb(NO_3_)_2_, Fe(NO_3_)_3_, Hg(NO_3_)_2_, and Cd(NO_3_)_2_ were used (all Aldrich salts at p.a. grade). KCl, NaNO_3_, Na_2_SO_4_, NaHSO_4_, Na_2_CO_3_, CH_3_COONa, KBr, NaNO_2_, Na_2_SO_3_, K_3_PO_4_, and NaF were the sources of anions (all salts at p.a. grade).

### 3.2. Methods

FT-IR spectra were recorded on a Thermo Scientific Nicolet iS20 FTIR spectrometer (Thermo Fisher Scientific, Waltham, MA, USA). The ^1^H NMR analysis was performed on a Bruker AV-600 spectrometer (BRUKER AVANCE II+ 600 MHz, Bruker, Billerica, MA, USA) with operating frequency at 600 MHz. Electrospray ionization mass spectra (ESI-MS) were obtained on a Bruker MicrOTOF-Q system (Compass, Bruker Billerica, MA, USA). The TLC monitoring was performed on silica gel, ALUGRAM^®^SIL G/UV254, 40 × 80 mm, 0.2 mm silica gel 60. A Hewlett-Packard 8452A spectrophotometer (Agilent Technologies, Inc., Santa Clara, CA, USA) was used for the UV-Vis absorption measurements. The photophysical study was performed at room temperature (25.0 °C) in 1 × 1 cm quartz cuvettes. The fluorescence spectra were recorded using a Scinco FS-2 spectrofluorimeter (Scinco, Seoul, Korea). The quantum yields of fluorescence (*Φ*_F_) were calculated relatively to Coumarin 6 (*Φ*_F_ = 0.78 in ethanol) [60]. Very small volumes of hydrochloric acid and sodium hydroxide were used to adjust the pH which was monitored by HANNA^®^ Instruments HI-2211 benchtop pH meter (HANNA Instruments, Woonsocket, RI, USA). The influence of metal cations and anions on the fluorescence emission was studied by adding portions of ion stock solution to a 10 mL of the fluorophore solution. The addition was limited to 100 μL so that dilution remained insignificant. The ions were added gradually up to 10 equivalents (10^−4^ mol/L) to a fluorophore solution (10^−5^ mol/L). The effect of ions was studied at constant pH in the presence of 10 µM HEPES (pH 7.2) or 10 µM Tris-HCl (pH 8) buffer solutions. The viscosity measurements were performed in binary mixtures of ethylene glycol/glycerol using an Ubbelohde capillary viscometer (Sigma-Aldrich Co., St. Louis, MO, USA).

### 3.3. Synthetic Procedures

#### 3.3.1. Synthesis *N*-Amino-4-chloro-1,8-naphthalimide **2**

A hydrazine monohydrate (0.2 mL, 400 mmol) was added to a solution of 4-chloro-1,8-naphthalic anhydride **1** (1 g, 400 mmol) in 20 mL of methanol. The resulting mixture was heated under reflux for 3 h. After cooling, the precipitate was filtered off, washed with methanol, and dried to produce pale yellow crystals of *N*-amino-4-chloro-1,8-naphthalimide **2** (0.77 g, 78%). FT-IR (KBr) cm^−1^: 3314 and 3236 (*ν* NH_2_); 1702 (*ν*^as^ N-C=O); 1652(*ν*^s^ N-C=O).

#### 3.3.2. Synthesis of Probe **3**

To a solution of *N*-amino-4-chloro-1,8-naphthalimide **2** (0.5 g, 2 mmol) in 5 mL of DMF, 0.4 mL of methylpiperazine (8 mmol) was added; then, the resulting solution was heated under reflux for 5 h. The precipitated solid after cooling was filtered off and dried. The final 1,8-naphthalimide **3** was obtained as yellow crystals (0.62 g, 99%) after azeotropic distillation of the solvent (DMF) in the presence of *n*-heptane. FT-IR (KBr) cm^−1^: 3331 and 3335 (νNH_2_); 1693 (ν^as^N-C=O); and 1633 (ν^s^N-C=O). ^1^H NMR (CHCl_3_-*d*, 600.13 MHz) ppm: 8.53 (dd, 1H, *J* = 7.3 Hz, *J* = 1.1 Hz, naphthalimide H-5); 8.46 (d, 1H, *J* = 8.1 Hz, naphthalimide H-2); 8.36 (dd, 1H, *J* = 8.4 Hz, *J* = 1.1 Hz, naphthalimide H-7); 7.63 (dd, 1H, *J* = 8.4 Hz, *J* = 7.3 Hz, naphthalimide H-6); 7.15 (d, 1H, *J* = 8.1 Hz, naphthalimide H-3); 5.44 (br.s, 2H, NH_2_); 3.26 (m, 4H, 2 × NCH_2_); 2.69 (m, 4H, 2 × CH_3_NCH_2_); and 2.38 (s, 3H, CH_3_). Elemental analysis: Calculated for C_17_H_18_N_4_O_2_ (MW 310.35) C 65.79, H 5.85, N 18.05%; Found C 66.01, H 5.79, N 17.88%. Positive-ion ESI-MS at *m*/*z*: 311.0145 [M + H]^+^.

## 4. Conclusions

In conclusion, a novel highly water-soluble 1,8-naphthalimide with pH- and viscosity-sensing fluorescence was designed using TICT molecular motion. The photophysical investigation of the synthesized compound was performed in an aqueous solution and in solvents with different viscosities. The results obtained revealed its potential to serve as a fluorescent probe for rapid detection of pH and viscosity, which was attributed to the destabilization of the TICT excited state in acid media and hindered rotation at high viscosity. The results presented here could be seen as a contribution to the development of the applied sensory chemistry using environmentally friendly aqueous media.

## Data Availability

The authors declare that the data supporting the findings of this study are available within the article.

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
