# Peer review of "A Novel Fluorescent Probe for Determination of pH and Viscosity Based on a Highly Water-Soluble 1,8-Naphthalimide Rotor"

_molecules, 2022, doi:10.3390/molecules27217556_

Round 1

Reviewer 1 Report

In this paper, a water-soluble 1,8-naphthalimide derivative was synthesized and applied as fluorescence sensor for pH and viscosity. Since similar results have already been reported in literatures, the novelty of this manuscript is limited. Publication of the manuscript could be considered only after the authors revise following issues.

1. The sensing performances for both pH and viscosity were measured for only once. Obviously, fluctuation is significant for the obtained spectra in Figures 2 and 3. Therefore, the authors should check the sensing performance with parallel tests and provide corresponding error bars in the analyte-signal relationship plots.

2. The report of the simplest “OR” logic gate may be not interest to readers. Especially, the response signal of viscosity is much lower than that of pH, making the “OR” gate much less attractive. If the authors could realize the “OR” gate operation in living systems such as cells, it will be much better results.

Author Response

1) The sensing performances for both pH and viscosity were measured for only once. Obviously, fluctuation is significant for the obtained spectra in Figures 2 and 3. Therefore, the authors should check the sensing performance with parallel tests and provide corresponding error bars in the analyte-signal relationship plots.

According to the recommendation of Reviewer #1, the sensing performance was check with three parallel tests and the observed standard deviations were included as error bars in the observed calibration plots. As a result Figure 2A and Figure 2B were replaced with new Figure 2A and Figure 2B.

2) The report of the simplest “OR” logic gate may be not interest to readers. Especially, the response signal of viscosity is much lower than that of pH, making the “OR” gate much less attractive. If the authors could realize the “OR” gate operation in living systems such as cells, it will be much better results.

We agree with the reviewer's recommendation that the study of “OR” gate operation in living systems would be more attractive and interesting to the readers. However such experimental work requires a study of cell culture in several months, which will be the subject of our future research.

Reviewer 2 Report

The manuscript reports a new lumiescent molecular rotor capable of sensing applications. The paper is well written and presents results of a wide general interest. However, some important issues need to be resolved before possible acceptance: 

1. Piperazine ring is a very flexible moiety which can undergo diverse deformations in the six-membered ring and N atom inversion besides the mentioned C-N bond rotation. Luminescence increase due to the blocking of TICT by the protonation of N atom looks intuitively resonable, however, at least excited state lifetimes could be measured to unambiguously confirm this statement. 

2. Fig. 2 presents very low intensities of the spectra. Any explanation? 

3. Glycerol has lower pKa value than glycol. Is it possible if acidity of glycerol/glycol mixture has a major impact on the spectra presented in Fig. 3? 

4. What about stability of 1,8-naphthalohydrazide core in the highly alkaline media? May its hydrolysis occur at pH=11-12? 

Author Response

1) Piperazine ring is a very flexible moiety which can undergo diverse deformations in the six-membered ring and N atom inversion besides the mentioned C-N bond rotation. Luminescence increase due to the blocking of TICT by the protonation of N atom looks intuitively resonable, however, at least excited state lifetimes could be measured to unambiguously confirm this statement.

The presence of the TICT process in the fluorophore excited state for dialkylamines, including N-methylpiperazine, is well known and confirmed in the literature. Destabilization of the TICT process that prevents rotation of the C-N bond by protonation is also well known in the literature (Section 2.1. Design and Synthesis). Our studies in a medium with increasing viscosity unequivocally confirm the presence of a TICT effect in the molecule, whose destabilization as a function of the viscosity and pH of the medium is the cause for the sensory properties of the new compound (2.2.2. Viscosity Sensing Properties).

2) Fig. 2 presents very low intensities of the spectra. Any explanation?

The measurements in Fig. 2 were performed in pure water solution which is well known as an effective fluorescent quencher.

3) Glycerol has lower pKa value than glycol. Is it possible if acidity of glycerol/glycol mixture has a major impact on the spectra presented in Fig. 3?

In order to reject this assumption the fluorescent spectra of 3 were measured in glycol, buffered glycol containing 10 µM Tris-HCl (pH 8), glycerol and buffered glycerol containing Tris-HCl (pH 8). The observed results showed that the presence of buffer solution didn’t lead to a significant result which clearly illustrated that the changes in fluorescence intensity of 3 mainly were induced due to the different viscosity.

4) What about stability of 1,8-naphthalohydrazide core in the highly alkaline media? May its hydrolysis occur at pH=11-12?

It was found that the studied probe 3 could be transferred between “off” and “on” states reversibly several times without changes in the fluorescence intensity in both acid and alkaline media. These results clearly showed that compound 3 is stable. 

Round 2

Reviewer 1 Report

I do not have any further comments. The authors have revised accordingly.

Reviewer 2 Report

Authors have carefully addressed most of the reviewer's notes and now the manuscript can be accepted for publication.